# Optimization of Inventory Management to Prevent Drug Shortages in the Hospital Supply Chain

**Tarek Abu Zwaida \*, Chuan Pham and Yvan Beauregard** 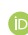

Synchromedia-École de Technologie Supérieure, Université du Québec, Montreal, QC H3C1K3, Canada;
chuan.pham@etsmtl.ca (C.P.); Yvan.Beauregard@etsmtl.ca (Y.B.)
**\*** Correspondence: tarek.abu-zwaida.1@ens.etsmtl.ca

**Abstract:** Drug shortage is always a critical issue of inventory management in healthcare systems since it potentially invokes several negative impacts. In supply chain management, optimization goes hand-in-hand with inventory control to address several issues of the supply, management, and use of drugs. However, it is difficult to determine a shortage situation in a hospital due to multiple unpredictable reasons, such as manufacturing problems, supply and demand issues, and raw material problems. To avoid the shortage problem in a hospital, efficient inventory management is required to operate the system in a sustainable way and maximize the profit of the organization in the Hospital Supply Chain (HSC). In this work, we study a drug refilling optimization problem, a general model for drug inventory management in a hospital. We then investigate a Deep Reinforcement Learning (DRL) model to address this problem under an online solution that can automatically make a drug refilling decision in order to prevent a drug shortage. We further present a numerical result to verify the performance of the proposed algorithm, which outperforms the baselines (e.g., over-provisioning, ski-rental, and max-min) in terms of the refilling cost and the shortage rate.

**Keywords:** hospital supply chain; drug shortage prevention; inventory management

## 1. Introduction

In the healthcare system, drug inventory management represents a significant portion of the costs, especially in the hospital supply chain in order to efficiently control and satisfy usage requirements [1]. However, the general healthcare system is faced with the challenges of increasing efficiency and reducing waste. First, the implication is that no healthcare facility is immune to drug shortages [2]. Most hospitals have experienced drug shortage or have been affected by this in their systems. Drug shortages are caused by many reasons and have forced healthcare organizations to purchase a more expensive alternative to operate their systems in a sustainable way. Second, an over-provisioning mechanism is often employed in the hospital as a solution, referred to as the safety stock [3]. The safety stock is a value that is calculated taking into account the variability of demand during the lead time, as well as the variability of delays in getting the ordered goods. Indeed, this method can mitigate the shortage situation in a hospital, but it often causes a high cost to buy and store medicines in the hospital since it is agnostic of the high costs of buying and maintaining them in the system. Specifically, a high imported volume of drugs without taking into account the user demand might also result in a high cost to store and prevent damage. Some drugs require special storing conditions, while most of have specific expiration dates and cannot be used after that.

An efficient inventory management is considered as a solution to improve the quality of customer service and the organization in the Hospital Supply Chain (HSC). Investigating the drug shortage problem, Reference [4] stated that there can still be nationwide drug shortages. In fact, the danger of a drug shortage can be caused in one area, but it might have a domino effect in other regions, even if they are not in the same part of the health system. However, a local optimal system can mitigate this negative impact. The creation

of clear lines of communication, gaining transparency into the cause of the shortages, and continually accessing in-stock and low-cost alternatives are the best mitigation process for developing a balanced inventory [5]. The Food and Drug Administration (FDA) has developed the Innovation Act and Strategic Plan to help mitigate drug shortages. The World Health Organization (WHO) has recognized drug shortage as a global problem [6]. Integrating manufacturers, suppliers, wholesalers, and stores to produce and distribute the right quantities of drugs at the right time in the right place to satisfy service-level requirements and minimize system-wide cost is called supply chain management.

Therefore, optimization is needed, and this goes hand-in-hand with the supply chain management of the inventory to address how medicines and drugs are supplied, managed, and used. Understanding the underlying causes of drug shortages is the most critical aspect that any hospital or health system needs to be acquainted with to help mitigate these outcomes [7]. By doing this, it is also easy to come up with proactive measures with the available resources. Determining the exact cause of drug shortages appears to be difficult; in general, it has been divided into three categories: manufacturing problems, supply and demand issues, and raw material problems [8]. Hence, in this study, we focus on the supply and demand issues, which means that given the information of the supply chain system, we aim to control the drug inventory based on the uncertainty of user demand and drug prices. Our model aims to make an automated decision in order to answer the following questions, which have seen little research: (i) *Should we refill drugs* ? (ii) *How much drug should be refilled*?

More precisely, this research aims to study an inventory drug optimization problem, which can capture the requirement constraints of drug demands, storage capacity, and refilling conditions while minimizing the refilling costs in the system. Since our designed problem is NP-hard, it is impossible to find a solution in polynomial time. To deal with this problem, we advocate a Markov Decision Process (MDP) to model the drug inventory system in order to address the shortage problem. Based on that model, we propose a Deep Reinforcement Learning (DRL) framework model that combines the Reinforcement Learning (RL) [9] method and the Deep Neural Network model(DNN) [10] to automatically make a decision in a finite horizon.

Basically, RL is modeled as an MDP that is comprised of three concepts: a state, an action corresponding to a state, and a reward for that action. Following the loop of actions and observations, the agent in an MDP often refers to a long-term consequence. Thus, RL is particularly well suited to control the drug inventory in a finite horizon. Furthermore, the combination of RL and DNN can figure out the strict requirement of the MDP regarding the exact knowledge of the state space of the MDP, and DRL is able to find a near optimal solution for a large MDP model, as shown in Dynamic Refilling dRug Optimization (DR2O). With a large number of drugs in the system, it might be unsolvable for a classical dynamic programming method.

All of the contributions of our work are summarized as follows:

- First, we investigate the supply chain model of drugs in a hospital to formulate an optimization drug shortage problem, named Dynamic Refilling dRug Optimization (DR2O). We model an objective function that aims to minimize the refilling costs, comprised of the costs to buy, to store medicines, and the penalty cost due to shortage. Furthermore, we consider supply constraints to refill drugs, such as storage capacity, and budget constraints.
- Second, we propose a deep learning method based on RL and DNN, named the Deep Reinforcement Learning model for Drug inventory (DRLD), where the situation of drugs is formed as a state in a Markov Decision Process (MDP) [9]. Depending on each state, we look for a suitable action to make a refilling decision in order to minimize the objective cost function. Based on the MDP model, we design an online method to control the system, where a reward and *Q*-matrices are built to evaluate an action corresponding to each state. Due to a large searching state space in RL, we introduce a

DNN model that can approximate the *Q*-values after training that is able to learn the behavior of the system.

- Finally, we consider an intensive simulation to conduct our work. In detail, we make a comparison between our method and three baseline approaches, including over-provisioning, ski-rental [11], and max-min [12]. Our method outperforms in most evaluations, especially in reducing the refilling cost and shortage situation.

The rest of our study is organized as follows. In Section 2, we discuss selected prior works that relate to our study. In Section 3, we present the problem formulation of the supply chain model of drugs in a hospital. Section 4 discusses the reinforcement learning framework to deal with DR2O. We then present the simulation result in Section 5. Finally, we conclude our work in Section 6.

## 2. Related Work

This section reviews the existing literature related to inventory management and how its optimization can be used to prevent drug shortages in the hospital supply chain. In order for this goal to be realized, we will commence by defining several terms that are associated with the topic: inventory management and Hospital Supply Chain (HSC). In addition, we aim to expound on the importance of optimizing inventory management to prevent drug shortages in the HSC. Ultimately, the section illustrates how healthcare institutions are minimizing drug shortage costs in their hospital supply chains.

### 2.1. Defining Inventory Management and the Hospital Supply Chain

In line with [13,14], inventory management is connected with the procedure of requesting, storing, and utilizing an institution's inventory. This involves the management of primary products, components, as well as end products. It also consists of warehousing and processing such items. However, Reference [15] depicted it as a systematic approach to sourcing, storing, and selling inventory, that is both finished goods (products) and raw materials (components). A supply chain, on the other hand, is composed of stages that are either indirectly or directly involved in accomplishing a customer's request [16]. According to [17], it basically involves the producer, supplier, transport operators, warehousing, retailers, third party logistics providers, and lastly, the customer. Moons et al. noted that the supply chain is responsible for ascertaining that there is an adequate connection of hospitals, operations, and the revenue cycle.

### 2.2. The Importance of Optimizing the Inventory Management to Prevent Drug Shortages of the Hospital Supply Chain

Health care institutions across the globe are in search of methods that will prove effective in improving the efficiency of operations, that is inventory management, while reducing expenditures that will in no way affect medical care and services [18]. Reference [19] illustrated that the material requirements for the provision of health care delivery are multifarious, generating a complex distribution network of relationships from the distributor to the customer. Furthermore, health care budgets are very stringent, and thus, health care providers are attempting to optimize their inventory management, which will eventually lead to a reduction of the costs incurred whilst providing health care [13].

Additionally, Kritchanchai et al. in [20] noted that an effective supply chain management is one that intends to optimize the full value created as opposed to the profit produced in a specific supply chain. The hospital supply chain, sometimes referred to as the Pharmaceutical Supply Chain (PSC), is intricate and comprised of numerous organizations that perform various but sometimes superimposed roles in the contraction and distribution of drugs [21]. In line with [22], price variation among the various types of users is considered to be a regular phenomenon owing to its degree of complexity. Thus, it becomes significantly more complicated for policy makers to evaluate and comprehend the supply chain [23]. According to [16], increased discernment of said issues

associated with policymakers is considered to play a role in making logical policy decisions for Medicare programs.

An ineffective hospital supply chain is associated with product shortages, product discontinuity, decreased patient safety, poor performance, distribution flaws, and technological mistakes that result in stock shortages in hospitals [24]. Reference [25] stated that increasing the productivity of supply chain management is key to obtaining more robust, safer, and lower cost hospital operations in public hospital medicine management units. This is achieved by optimizing the supply processes, enhancing satisfaction and patient safety, as well as reducing errors. However, Michigan State University [26] observed that an extensive distribution arrangement to transfer pharmaceutical commodities and other medical equipment from the medical stores to the service points is still lacking in most hospital systems. This lack of appropriate distribution systems creates a considerable bottleneck, often making it very difficult to access said products and supplies [21].

### 2.3. Minimization of Drug Shortages in Hospital Supply Chains

The Canadian pharmaceutical supply chain, consisting of governments, manufacturers, wholesalers, distributors, pharmacists, and physicians, acknowledges the significance of a reliable medication supply [27] Moreover, it is obvious that there is insufficient information for accessing the drugs due to different issues, for instance the distribution system of the manufacturing [28]. However, there are instances when drugs do not reach the intended locations due to glitches in the distribution system. In instances where clients cannot access important drug products, practitioners have the responsibility of knowing the reasons for the product's unavailability, the time when the product is available, available alternatives to the unavailable drug product and the involved costs, how to obtain the unavailable product from alternative sources, and additional information detailing patients' needs and healthcare providers' needs. However, it is worth noting that it is unacceptable for hospitals to experience drug shortages [29], along with the associated costs.

There are various factors that could cause pharmaceutical supply disruptions, and they include unexpected increased demands in the drug's utilization, leading to an impermanent shortage. Such a shortage ends only when the manufacturing capability increases to a level that meets the identified demand. Pharmaceutical supply disruptions are also caused when products are voluntarily recalled or discontinued by a manufacturer. Disruptions also occur when Health Canada withdraws drugs from the market or when natural disasters such as floods and storms occur. These causes are responsible for the shortage costs experienced in the hospital supply chain inventory management. One needs to understand that the costs emanate from the fact that a client could not access the required drug product at the expected time. Additionally, the costs also include the time taken to purchase the drug product from alternative sources or acquiring alternative medications representing the unavailable drug product. The inventory process needs to be planned effectively to ensure that even in instances of unpredictable natural calamities, the duration of drug shortage is minimized [30].

Improper medical inventory management and drug shortages severely disrupt the HSC, which eventually leads to health services that are impoverished while increasing costs [22]. Kees et al. in [23] elucidated that many hospitals and pharmacies are subject to numerous problems as they aim to achieve proper inventory control. This is on account of the fact that they have hardly addressed how medicinal products are administered, supplied, and utilized to improve health, as well as to save lives. An additional crucial issue in the HSC as illustrated by [31] is unidentified occurrences. Such disasters generate enormous losses. In order to enhance the independence of an HSC from disasters, Reference [32] proposed an inventory control model that utilizes mathematical programming methods. The suggested model takes into account multiple forms of medications, ordering size, processing time, expiration dates of products, and customer service level, not to mention the holding and storage cost [33]. The equivalent answer or solution ascertains the optimum period of preparation and inventory level in the disaster formulation phase

with the lowest overall cost. However, Reference [20] elucidated that this method permits a handful of measuring points to distort the prediction and neglects accounting for seasonal changes and other variables. Thus, hospitals are left exposed to high costs and avoidable waste. Bradely et al. in [19] supported this argument illustrating that mathematical and statistical methods are proving to be inadequate to optimize the HSC.

Additionally, Cardinal Health [34] has found that manual processes and workflows in medication inventory management are not just slow and troublesome, but are actually insufficient at collecting intelligent information. According to [35], when a hospital's personnel is entrusted with the task of manually monitoring, logging, and restocking the inventory, the results are either imprecise, as human errors are bound to happen, or insufficient whenever the proper processes were not followed to the letter. Hence, as mentioned by [36], the use of Machine Learning (ML), which is a type of algorithmic training (or Artificial Intelligence (AI)), where the algorithms are constantly processed as additional data enables them to be more predictive, is gaining momentum in HSC management. Wild showed how ML can help HSC management to become more refined while making it less cumbersome [18]. By swiftly processing enormous volumes of data in order to discern patterns and uncover insights that may be too complex for or hidden from human perception (even those with considerable experience), ML has the capacity to enable health care providers to steadily provide the right provisions, at the right cost, place, and time [37].

Drug shortages can also be caused by increased demand or decreased supply [38]. Increased demand, particularly where the parties in the pharmaceutical supply chain employ just-in-time inventory control [38], has been found to be the cause of shortages close to 13% of the time [39]. That same study found that drug shortages caused by decreased supply due to manufacturing problems, at about 23%, surpasses drug shortages caused by increased demand [39].

Not only are genuine real-time computerized systems utilized to streamline the performance of tasks for medication inventory management, they also present a historical reporting that is accurate, complete, and real-time [36]. This is achieved through RL. This method is considered to be a subset of ML consisting of adopting appropriate actions so as to maximize the rewards in a certain condition [32]. However, Reference [37] illustrated that when an instruction dataset lacks information, it is bound to ascertain the missing information from its past experience. Pharmacy staff and purchasing managers utilize this information in order to optimize inventory so as to prevent drug shortages, as well as to give priority to patient safety while cutting unnecessary costs [33].

The mathematical programming model can be used in the determination of automated refilling decisions. For example, a mathematical model considers the various types of drugs, ordering sizes, shortages, and holding costs. Therefore, a hospital can use the model to compute these aspects, as well as determine the specific refilling schedule of drugs in supply. If there is a drug shortage, the costs caused by the shortage can be automatically anticipated to determine the necessary amount of drug refillings. To the best of our knowledge, the studies presented to date do not have a learning model to deal with the shortage issue. Therefore, in this work, we propose a deep learning model to formulate and develop a dynamic inventory model for drugs in a hospital setting.

## 3. System Model

We consider an enterprise supply chain to provide drugs to a hospital. This supply chain consists of a set of drugs $\mathcal{I}$ managed during period $T$. To simplify the problem formulation, all the notations are presented in Table 1. We assume that the system works in a time-slotted fashion, spanning time slots $1, 2, \ldots, T$, with an index $t$ that corresponds to a week or a month defined by the hospital.

Each drug $i \in \mathcal{I}$ has a storage capacity $C_i(t)$ and an expired function $e_i(t)$ that is used to measure the amount of expired drug $i$ at time $t$. With different types of drugs, the storage capacity units could be different (e.g., box, bottle, etc.) so that our model could flexibly

define the capacity unit for each specific drug $i$. To make a refilling decision, we consider a refilling cost $p_i(t)$, which can be measured or collected by prediction methods. In this work, we simply model this function by the regular (or base) price $\underline{p}_i(t)$ and the penalty price $\beta_i(t)$ due to the emergency demand. The function is calculated as follows:

$$p_i(t) = \underline{p}_i(t) + \beta_i(t), \forall i \in I, t = 1, 2, \ldots, T. \tag{1}$$

Practically, we are unaware of the price in advance and, therefore, cannot make a long-term plan. In this work, we presume that the price parameters are given before making a refilling decision at time $t$. We also denote the budget $B_i$ that is used to buy drug $i$ during time $T$.

**Table 1.** Notations.

| Symbols | Description |
|---|---|
| $\mathcal{I}$ | Set of drugs |
| $i$ | Drug index. |
| $t$ | Time index. |
| $T$ | Spanning time. |
| $e_i(t)$ | An amount of expired drug $i$ at time $t$. |
| $p_i(t)$ | The refilling cost of drug $i$ at time $t$. |
| $\psi(.)$ | The penalty function of drug $i$. |
| $\underline{p}_i(t)$ | The base price of drug $i$ at time $t$. |
| $\beta_i(t)$ | The penalty price of drug $i$ with an emergency demand. |
| $B_i$ | The budget of drug $i$. |
| $\mathbb{B}$ | The total budget. |
| $\alpha_i$ | The weight parameters of drug $i$. |
| $R_i(.)$ | The storage cost function of drug $i$. |
| $r_i(t)$ | The storage cost of drug $i$ at time $t$. |
| $\lambda_i(t)$ | The amount of demand of drug $i$ at time $t$. |
| $e_i(t)$ | The amount expired at time $t$. |
| $v_i(t)$ | The remaining volume of drug $i$ at time $t$. |
| $C_i(t)$ | The storage capacity of drug $i$. |
| $\underline{C}_i(t)$ | The lowest volume requirement of drug $i$. |
| $\underline{\rho}_i(t)$ and $\bar{\rho}_i(t)$ | The upper and lower bound refilling of drug $i$ at time $t$. |
| **Variables** | |
| $\boldsymbol{x}$ | The decision refilling volume variable. |
| $x_i(t)$ | The refilling volume of drug $i$. |

*3.1. Problem Formulation*

We aim to design an online solution that dynamically determines when to refill drugs over the entire running span $T$ while satisfying all requirements with a minimum refilling cost. Specifically, we aim to avoid a shortage situation in the system, which could lead to a domino effect. There is a general trade-off between cost optimization and any shortages due to unknown patient demand: by refilling more drugs, the shortage situation may be avoided, but the cost will be increased, and vice versa. In this work, we combine these dual goals in the objective function by using weighted parameters to weight the priority of each aspect.

We design a main variable $\boldsymbol{x} = \{x_i(t)\}_{\forall i \in \mathcal{I}, t=1,2,\ldots,T}$ to represent the amount of drug $i$ to refill at time $t$. The refilling cost model in this work contains the following parts:

- **Medicine cost:** We consider the first term as the medicine cost depending on the amount of drugs being ordered: $\sum_{t=1}^{T} \sum_{i \in \mathcal{I}} \alpha_i p_i(t) x_i(t)$, where $\alpha_i \in (0, 1]$ is the weight parameter of drug $i$ and $p_i(t)$ is the amount of money required to purchase the required

amount of drug $i$. Depending on the drugs' priorities, $\alpha_i$ is set with a high or low value.

- **Storage cost:** We design the second term as the storage cost incurred when the hospital stores drugs. Hospitals may incur a variety of costs associated with safely storing medicine, and some must utilize third-party storage. In this work, we generally model the storage cost as a function $R_i(v_i(t), r_i(t))$ depending on $v_i(t)$, the amount of drugs, and $r_i(t)$, the storage cost. The volume of drug $i$ at time $t$ can be obtained by:

$$v_i(t) = v_i(t-1) - \lambda_i(t) - e_i(t) + x_i(t), \tag{2}$$

where $v_i(t-1)$ is the remaining volume of drug $i$ at timeslot $t-1$, $\lambda_i(t)$ is the amount of the demand, and $e_i(t)$ is the amount of the drug available before expiration at time $t$.

- **Penalty cost:** A penalty cost $\psi(v_i(t))$ is assigned to prevent a shortage situation in the system. This cost depends on the volume and the minimum requirement for the drugs at time $t$. Without loss of generality, we refer this cost to a quadratic function to formulate our model. As shown in Figure 1, the quadratic penalty cost increases when the volume of drug $i$ reaches the bounded values $\underline{C}_i$ and $\bar{C}_i$. In other words, the penalty cost function aims to prevent both the shortage and overstock problems.

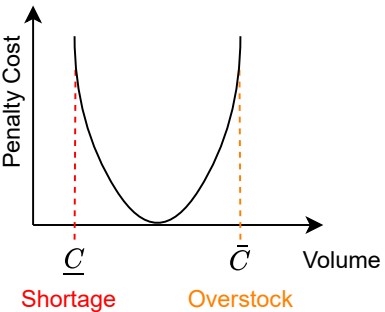

**Figure 1.** A quadratic function of the penalty cost.

Consequently, the objective function in our model is to minimize the cost incurred when the hospital makes a refilling decision. The objective function is expressed as follows:

$$\min \sum_{t=1}^{T} \sum_{i \in \mathcal{I}} \alpha_i p_i(t) x_i(t) + R_i(v_i(t), r_i(t)) + \psi(v_i(t)) \tag{3}$$

We next formulate the set of constraints that the decision variables should respect. First, we consider the capacity constraint to ensure that the refilling decision will not exceed its storage capacity $C_i(t)$. Furthermore, the system has to ensure the lowest volume $\underline{C}_i(t)$ of each drug $i$ to avoid a shortage. Hence, we formulate the constraint as follows:

$$\underline{C}_i(t) \leq v_i(t) = v_i(t-1) - \lambda_i(t) - e_i(t) + x_i(t) \leq C_i(t), \forall i \in \mathcal{I}, t = 1, 2, \ldots, T. \tag{4}$$

Second, we consider the budget constraints of the hospital for purchasing drugs during time $T$ as follows:

$$\sum_{t=1}^{T} \sum_{i \in \mathcal{I}} x_i(t) p_i(t) \leq \mathbb{B}. \tag{5}$$

In addition, there is a budget for each medicine $i$ at time $t$, which is considered based on the following constraint:

$$x_i(t) p_i(t) \leq B_i(t), \forall i \in \mathcal{I}. \tag{6}$$

Finally, at each timeslot $t$, the hospital could have a specific refilling range for each drug $i$ in order to provide for a specific requirement in the hospital. We formulate this constraint as follows:

$$\underline{\rho}_i(t) \le x_i(t) \le \bar{\rho}_i(t), \forall i \in \mathcal{I}, t = 1, 2, \ldots, T, \tag{7}$$

where $\underline{\rho}_i(t)$ and $\bar{\rho}_i(t)$ are the lower and upper bound refilling volumes, respectively, of drug $i$ at time $t$.

### 3.2. A Dynamic Refilling Drug Optimization Model

Based on the aforementioned objective and constraints, the Dynamic dRug Refilling Optimization (DR2O) model is presented as follows:

$$
\begin{aligned}
\min_{\mathbf{x}} \quad & \sum_{t=1}^{T} \sum_{i \in \mathcal{I}} \alpha_i p_i(t) x_i(t) + R_i(v_i(t), r_i(t)) + \psi(v_i(t)) \\
\text{s.t} \quad & \underline{C}_i(t) \le v_i(t) = v_i(t-1) - \lambda_i(t) - e_i(t) + x_i(t) \le C_i(t), \forall i \in \mathcal{I}, t = 1, 2, \ldots, T, \\
& \sum_{t=1}^{T} \sum_{i \in \mathcal{I}} x_i(t) p_i(t) \le \mathbb{B}, \\
& x_i(t) p_i(t) \le B_i(t), \forall i \in \mathcal{I}, \\
& \underline{\rho}_i(t) \le x_i(t) \le \bar{\rho}_i(t), \forall i \in \mathcal{I}, t = 1, 2, \ldots, T.
\end{aligned}
$$

In general, this problem can be seen as a variant of the classic ski-rental problem [11]. Specifically, a drug $i$ is similar to the person in the classic ski-rental problem who is going to ski, but does not know how long the snow will last. He/she therefore has to make a decision every day as to whether to buy (so he/she will not need to pay for the next days) or to rent skis to minimize the overall cost. In our problem, we have to decide whether to buy a drug or not at time $t$, and we do not know the user demand in advance, which could result in a shortage in the next period. A hospital might spend a good portion of its budget buying and storing a high volume of medicines that may not be used until their expiration. However, a hospital could fall into a shortage crisis if it manages a low supply of drugs. To address this problem, we advocate a learning method to deal with the following issues: (i) the uncertainty of demand that affects the decisions at every timeslot and (ii) the automation mechanism that can automatically make a decision to adapt to the environmental situation.

**Theorem 1.** *DR2O is NP-hard.*

**Proof.** Consider a knapsack problem with $N$ items where each item has a non-negative weight $w_i$ and a value $p_i$. There is a bound $W$ to select a subset $S$ of items where $\sum_{i \in S} w_i \le W$. The objective is to select a subset of maximum total value $\sum_{i \in S} p_i$, subject to the boundary constraint. Using a binary variable $x_n$ to indicate item $n$ is selected in the subset $S$, the problem formulation of the knapsack problem is as follows:

$$\max \qquad f(\mathbf{x}) = \sum_{n \in N} p_n x_n \tag{8}$$

$$\text{s.t.} \qquad \sum_{n \in N} x_n w_n \le W, \tag{9}$$

$$x_n \in \{0, 1\}. \tag{10}$$

Let us simplify the system by considering an offline model of DR2O, as using DR2O with an online form is always more complicated due to uncertain demands. This means that we can know exactly the amount of drugs used in timeslot $t$. Hence, as shown in Figure 2, to refill drugs in period $T$, we need to find a subset of the blocks presented for the amount of drugs in each timeslot $t$ in which the total is bounded by the budget $\mathbb{B}$. If we consider the revenue of each drug determined from the surplus between the value $\vartheta_i$ and

the refilling cost, the objective of DR2O can be formed similarly to the knapsack problem by maximizing the total revenue of all the drugs.

$$\max \sum_{t=1}^{T} \sum_{i \in \mathcal{I}} \vartheta_i(v_i(t-1) + \lambda_i(t)) - [\alpha_i p_i(t)x_i(t) + R_i(v_i(t), r_i(t)) + \psi(v_i(t))]. \quad (11)$$

Since the first term of (11) is constant because $\lambda_i(t)$ is given, this objective function can be maximized by minimizing the second term as we formulated in DR2O. Thus, the offline DR2O constructs an instance of the knapsack problem, which is proven to be NP-hard [40]. □

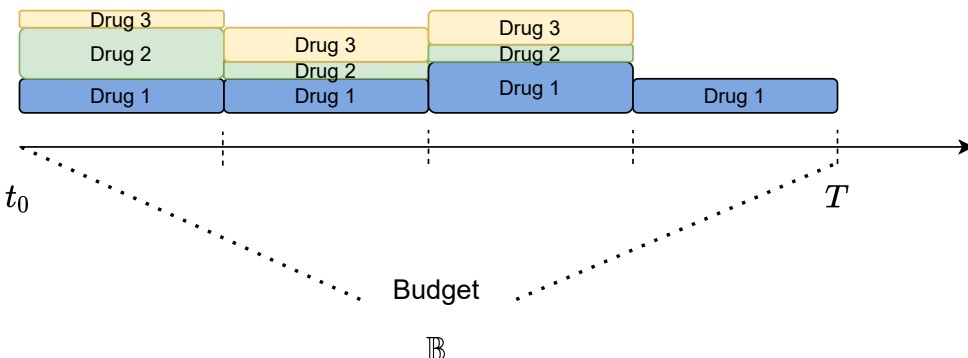

**Figure 2.** An example of the offline Dynamic Refilling dRug Optimization (DR2O).

## 4. Reinforcement Learning

### 4.1. Markov Decision Process Model

Firstly, we present the Markov Decision Process (MDP) to formulate our problem. Based on this model, we then propose a deep Q-network algorithm to find a solution for DR2O. In general, the MDP model is comprised of three concepts: a state, an action corresponding to a state, and a reward for that action.

**The system state** $\mathcal{S}$: We consider the system state at time instant $t$, including the current volume of drugs, the user demand, and the costs of buying and storing medicine. In addition, due to specific requirements at time $t$, some other information could be changed, such as the purchasing budget, the storage capacity for drug $i$, and the minimum and maximum amount of drugs to refill. For the set $\mathcal{I}$ of drugs $i$, we could make an order to make a decision for the drugs that have higher priority and then consider the remaining budget for the rest. Therefore, we make a loop with $|\mathcal{I}|$ iterations to make a decision for each drug $i$ in turn. We denote the state $s_i(t) = < p_i(t), r_i(t), \lambda_i(t), B_i(t), e_i(t), \underline{C}_i(t), C(t), \underline{\rho}_i(t), \bar{\rho}_i(t) >$ of drug $i$ at time $t$.

**Action set** $\mathcal{A}$: The action set in our model reflects the refilling decision of each drug $i$. In particular, the action $a(t) = \{a_i(t)\}_{\forall i \in \mathcal{I}}$, where $a_i(t)$ is the decision of drug $i$ at time $t$ that responds to the state $s_i(t)$.

**Reward** $W$: As presented for the objective function, our model aims to minimize the refilling cost including the purchasing and storing costs. Therefore, we first model the reward function based on the purchasing cost as follows $\frac{1}{L_i(x_i(t))}$, where $L_i^{(1)}(x_i(t)) = \alpha_i p_i(t)x_i(t) + R_i(v_i(t), r_i(t)) + \psi(v_i(t))$, and $k$ is the considered timeslot. This means that the more money the system needs for its purchases, the less reward it has. The function attempts to navigate the system, so it selects actions that obtain higher rewards, which is equivalent to minimizing the purchasing cost.

The next term involves the penalty for the shortage situation, as we aim to avoid this problem for all the drugs in storage. The penalty term is defined by $\psi(v_i(t))$ as formulated above.

We combine these terms in the penalty function by:

$$w_i(t) = (1 - \theta)\frac{1}{L_i(x_i(t))} + \theta\psi(v_i(t)), \tag{12}$$

where $\theta \in (0, 1]$ is the weighted parameter that is designed to set the priority of each term.

Considering duration $T$, the state of each drug and its reward are stochastic and follow the MDP, where the state $s_i(t)$ changes to $s_i(t+1)$ with a transition probability, and the reward depends on the state and the selection action.

To go from $s_i(t)$ to $s_i(t+1)$ with reward $w_i(t)$, we consider the conditional transition probability, $p(s_i(t+1), w_i(t)|s_i(t), a_i(t))$. It should be noted that the agent can only control its own actions and has no prior knowledge about the transition probability matrix $P = p(s_i(t+1), w_i(t)|s_i(t), a_i(t))$, which is determined by the environment. The intuition of the MDP is presented in Figure 3. Therefore, the main objective of this reinforcement learning is to find a policy to maximize the expected cumulative reward. We have:

$$R_i = E\Big[\sum_{t=k}^{T} w_i(t)\Big] \tag{13}$$

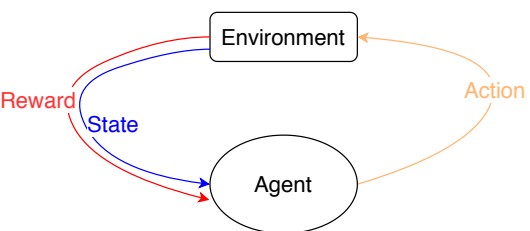

**Figure 3.** Markov decision process.

### 4.2. Deep Q-Learning

As shown in Figure 3, the agent takes actions depending on the state, called the policy $\pi$, which maps the state $s_i(t)$ to the action $a_i(t)$. Mathematically, we express this as $\pi_i : s_i(t) \in \mathcal{S} \to a_i(t) \in \mathcal{A}$. The Q-learning mechanism is used to maximize the long-term expected accumulated discounted rewards [10]. Considering the $Q_i(s_i(t), a_i(t))$ value of $\pi_i$ for a state $s_i(t)$ and action $a_i(t)$ pair, this value is calculated by the expected accumulated discounted rewards. Therefore, the policy $\pi_i$ is constructed by taking the action:

$$a_i(t) = \arg\max_{a \in A} Q_i(s_i(t), a_i(t)), \forall i \in \mathcal{I}. \tag{14}$$

Based on the Q-function from the Bellman equation [10], the optimal policy $\pi_i$ with value $Q_i$ can be obtained by:

$$Q_i(s_i(t), a_i(t)) = E\big[w_i(t+1) + \psi w_i(t+2) + \psi^2 w_i(t+3) + \ldots |s_i(t), a_i(t)\big] \tag{15}$$

so that the $Q$-value for the state, given a particular state, is the expected discounted cumulative reward.

Therefore, in the MDP, we aim to determine an optimal policy expressed as $\pi^* : \mathcal{S}_i \to \mathcal{A}_i$. Following the stationary distribution in the MDP, the Q-values will converge to the optimal value $Q^*$ with the following equivalent calculation [10]:

$$Q_i(s_i(t), a_i(t)) = Q_i'(s_i(t), a_i(t)) + \psi[w_i(t+1) + \delta\max_{s_i \in \mathcal{S}_\rangle} Q_i'(s_i(t), a_i(t)) - Q_i'(s_i(t), a_i(t))], \tag{16}$$

where $Q_i'(.)$ is the old value. To obtain the optimal $Q$-value, the algorithm is executed until the mean changed values of the $Q$-values is less than a threshold, called the training phase.

The details of the training phase can be described as shown in Algorithm 1. At the beginning, a random $Q$-matrix is generated (Line 3). A loop is executed (from Lines

5 to 9) to modify this matrix until convergence (i.e., where the change is less than the threshold $\epsilon$). A random state $s_i(t)$ is selected to start the training (Line 4). Line 6 randomly selects a possible state $a_i(t)$ to move to the next state $s_i'(t)$. Line 7 updates the $Q$-value of the state-action pair $(s_i(t), a_i(t))$ using (16). The algorithm continues until it meets the convergence condition.

---

**Algorithm 1:** Reinforcement learning.

---

**1** Input: $\mathcal{S}$ and $\mathcal{A}$;
**2** Output: $Q$-matrix;
**3** Initialization: $Q$-matrix;
**4** Select a random starting state $s_i(t)$ that has some possible actions from $\mathcal{A}$;
**5** **while** $||Q - Q'|| \leq \epsilon$ **do**
**6**   Select one of the possible actions $a_i(t)$ that moves to the next state $s_i'(t)$;
**7**   Update the $Q$-value of the state-action pair $(s_i(t), a_i(t))$ according to (16), then set to $Q'$;
**8**   Set $s_i(t) \leftarrow s_i'(t)$;
**9**   Go to Step 5;
**10** **end**

---

According to the traditional $Q$-learning method, it is not difficult to obtain the convergence result with a small state-action space. However, the classic $Q$-learning model cannot be applied directly to our work because the state-action space is huge, given that the state in DR2O is comprised of a tuple of parameters that generate a huge combination. Furthermore, the level of drug refilling $x_i$ is set flexibly in a specific range, and so, it also increases the size of the state-action space. In this case, there are two problems: (i) it is too difficult to build the transition probability for the MDP, and (ii) some states that are not visited and that are updated infrequently lead to a long convergence in the training phase to obtain the $Q^*$-value.

To deal with this problem, a Deep Neural Network (DNN) is used to approximate the $Q$-function [9]. Given the input information of a state $s_i(T)$, the DNN is trained to learn an optimal mapping $s_i(t)$ to $a_i(t)$. Therefore, we design the input of the DNN to present all the features of a state $s_i(t)$, and the output is the $Q$-values, $Q_i(s_i(t), a_i(t))$. We design this DNN model for the set of drugs $I$; therefore, for a simpler formulation, we remove the $i$ index.

To obtain the correct $Q$-values, the DNN needs a training phase to update the weight parameters in the network. Specifically, given an input-output pair $< s(t), y >$ in the data set $\mathcal{D}$, the DNN aims to minimize the following loss function:

$$L = \sum_{(s(t),y) \in \mathcal{D}} (y - Q(s(t), a(t)))^2, \tag{17}$$

and the given output $y$ is calculated by:

$$y = w(t) + \max_{a(t) \in \mathcal{A}} (Q(s(t), a(t))). \tag{18}$$

*4.3. Training and Testing Phases*

We present the training algorithm in Algorithm 2, which is called the Deep Reinforcement Learning approach for Drug inventory (DRLD). Similar to the standard $Q$-learning algorithm, an action of DRLD is selected based on the environment and the reward. Instead of using a full history as the $Q$-learning method to take an action for a current state, we limit the length of history defined by $\gamma(.)$, which is related to the number of input nodes in the DNN. In the training phase, we use $\epsilon$-sampling to generate and collect the data to train the weight parameter of the DNN.

The details of the algorithm are as follows. At the beginning, we initialize the current simulation environment with given sets of $\mathcal{S}$ and $\mathcal{A}$ (Line 1) and initialize for the learning

sequence set $\mathcal{D}$ and the DNN model (Lines 3-4). We select a random starting state $s(0)$ from the current $\mathcal{S}$ (Line 6) to execute $M$ training times. An action $a(t)$ is selected with an $\epsilon-$probability; otherwise, an action that maximizes $Q(s(t), a(t))$ is selected (Line 8). According to the action $a(t)$, the environment is set by a new state $s(t+1)$ with a reward $r(t)$ (Line 9). If the state $S(t+1)$ does not exist in $\mathcal{S}$, it will be added for training (Line 10). Furthermore, $\mathcal{D}$ is appended by the transition $s(t) \leftarrow s(t+1)$ (Line 11). The training in the DNN is started by sampling a sequence $s(j) \rightarrow s(j+1)$ in $\mathcal{D}$ (Line 12). The training phase is designed to update the weight parameters in DNN with the given state $s(j)$ and the output that is calculated by (18) (Lines 13–14). Note that the values of $M$ and $T$ are defined by our experience (e.g., $M = 7$ and $T = 30$). These values are changeable and can be observed depending on each system. However, technically, while setting the values of these parameters higher will increase the accuracy, in fact, it will result in a long convergence since it adds more iterations.

In the testing phase, the trained agent will select an action $a(t)$ with the maximum $Q$-value given by the training phase. Based on the training $Q$-matrix, we can operate the drug refilling model, DR2O, as an online mechanism.

---

**Algorithm 2:** DRLD- training algorithm.

---

1 Input: Start the environments $\mathcal{S}$ and $\mathcal{A}$;
2 Output: $Q$-matrix;
3 Initialization: $Q$-matrix, DNN model;
4 Set empty for the learning sequence $\mathcal{D}$;
5 **for** *j=1:M* **do**
6 　Select a random starting state $s(0)$;
7 　**for** *t=0:T-1* **do**
8 　　Sample an action $a(t)$ with probability $\epsilon$; otherwise, select $a(t) = \arg\max Q(s(t), a(t))$;
9 　　Generate a next state $s(t+1)$ and reward $r(t)$;
10 　　Add $\mathcal{S} = \mathcal{S} \cup s(t)$;
11 　　Add the transition $s(t) \rightarrow s(t+1)$ to $\gamma(s(0) \rightarrow s(t+1))$, and save in $\mathcal{D}$;
12 　　Sample a transition $\gamma_j(s(j) \rightarrow s(j'))$ from $\mathcal{D}$;
13 　　Set the output $y(k)$ by (18) to train the DNN;
14 　　Using a gradient descent to update weights of the DNN;
15 　**end**
16 **end**

---

### 4.4. Discussion

As presented in Algorithms 1 and 2, the action of each drug $i$ is chosen independently based on its environment. There is an issue in our system when the actions are executed simultaneously. In this case, the agent will not have the correct information about the environment, which is affected by the other action. To deal with this problem, we suppose that the action updates are executed asynchronously. For example, we rank drugs based on the priority of refilling to perform refilling action in a sequence. Hence, only one or a small subset of drugs will update their environments. This approach allows the environment changes caused by other agents to be observed accurately. In the real system, this modification of our mechanism is needed to coordinate the different drugs in the system.

### 5. Experiment and Numerical Results

#### 5.1. Experiment Configuration

In this section, we present our experimental evaluations, which mainly focus on the training time required for using reinforcement learning to obtain a near-optimal solution. We use the hardware configuration of the experimental environment with a CPU, 2.4 GHz Intel Core i5, and an 8 GB memory at 1600 MHz DDR3. Our simulator program was developed in the PyTorch framework [41] and the Ipot optimization library [42]. Due to the

difficulty of accessing real databases in a hospital, which contain very sensitive information, we refer to the data used in [43] with a list of 70 drugs. The setting range of the parameters in our work is shown in Table 2.

**Table 2.** Simulation settings.

| Parameters | Settings |
|---|---|
| Storage demand of drugs | [0.001–0.005] ft$^3$ |
| Total inventory capacity | 40 ft$^3$ |
| Refilling cost of drugs | [5–100] USD/unit |
| Storing cost | [2–5] USD/ft$^3$ |
| Drug demand | [5–20] unit |

In order to evaluate the system, we compare our work with the following baseline approaches:

- Optimal: We use the Ipopt solver [42] to solve the DR2O problem with the assumption that the demand during $T$ is given. Hence, the optimal result of DR2O, in this case, can be considered as the offline optimal solution to compare with other approaches.
- Over-provisioning: This is a simple strategy that is often carried out in the inventory system. Based on the average utilization (gathered from the log files of history), the expired period, and the current remaining volume of drugs, a refilling decision is considered. To prevent a shortage problem, the amount of drugs to be refilled is often provided with an additional volume, which results in a higher cost of operation.
- Ski-rental: As presented in the formulation, DR2O can be an instance of the ski-rental problem. Therefore, to evaluate the performance of the DRLD, we implement an online algorithm with the $c$-competitive value where $c = (2 - 1/T)$.
- Max-min [12]: This refilling strategy is one of the most useful mechanisms for inventory management. By using the min level, a trigger will be active to make a refilling decision to obtain the maximum target quantity. To prevent the shortage problem, the min value is often set with a high volume (we use 45% in our work); therefore, the max-min baseline also results in a high total cost.

*5.2. Results*

5.2.1. Convergence Evaluation

We first evaluate the convergence of our proposed method to illustrate its performance. As shown in Figure 4a, we show the reward values, which are significant because they impact the convergence, using different settings (e.g., datasets of 30, 50, and 70 drugs). With the smallest setting, we obtain the fastest convergence, in close to 2800 iterations, because this setting occupies the smallest state space. The system meets the stop condition at around 3200 and 4000 iterations for the larger settings of 50 and 70 drugs, respectively. This evaluation also illustrates a promising aspect of our proposed method: when we increase the number of drugs in the system, the number of iterations does not exponentially increase.

In the next evaluation, we show the mean error rate of the DNN in Figure 4b. This figure shows how the convergence of the DNN is affected by varying the learning rate. The DNN can reach a fast convergence when the learning rate is 0.01 (less than 5000 s), but the error rate is still high. After some experiments, we selected a learning rate of 0.005, which resulted in a low error rate where the system reached convergence after 10,000 s. Although it requires a long training phase, with this rate, the DNN has a better decision-making performance. The convergence status allows us to demonstrate the total cost in the system compared to the optimal results calculated by the Ipopt solver. In Figure 4c, a small gap is visible between the optimal value and the DRLD result for all settings. However, the optimal gap depends on the training phase and the training data. With our limited data set, we present a simple, but promising result in this work. We will investigate this aspect in future work to find a real public dataset with which to obtain a practical training model.

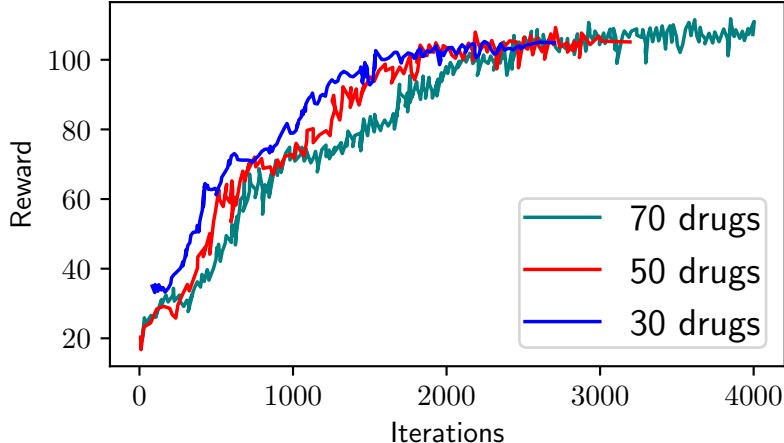

(**a**) Evaluation of the reward value.

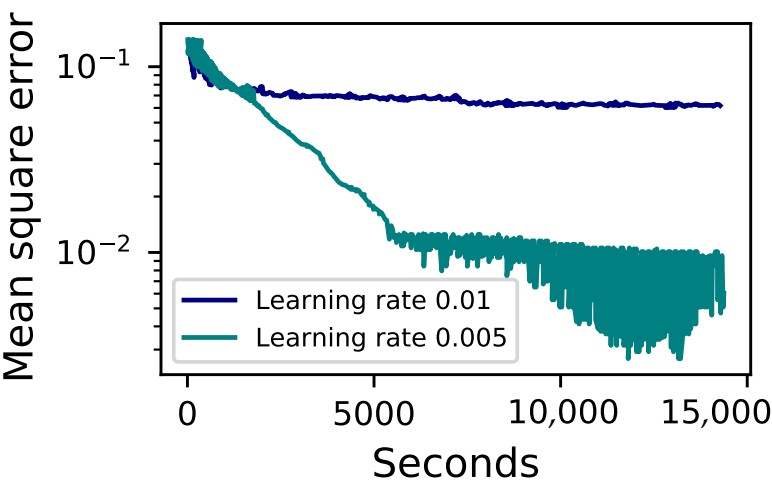

(**b**) Evaluation of the DNN.

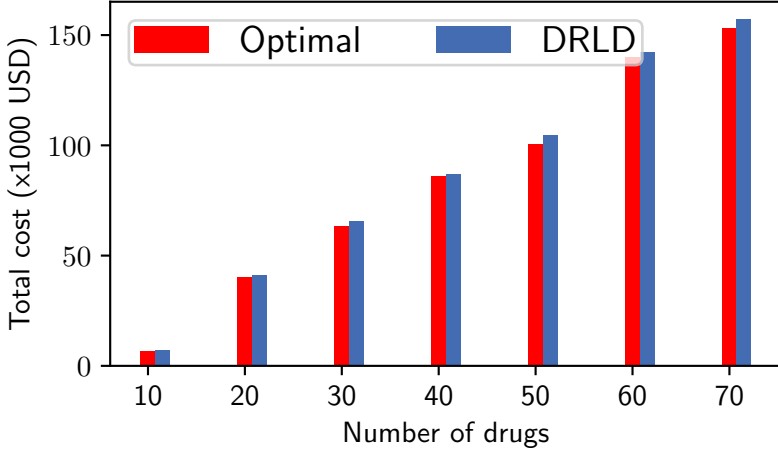

(**c**) Evaluation of the total cost.

**Figure 4.** Convergence evaluation. DRLD, Deep Reinforcement Learning model for Drug inventory.

5.2.2. Evaluation of the Refilling Cost

We consider the refilling cost in a time horizon with 30 timeslots (we refer to one week for each timeslot) as shown in Figure 5. We first make a comparison with a simple, commonly-used method, over-provisioning. Since the remaining volumes and the upper bound volumes of drugs are often used as the replenishment policy for each refilling inventory, they lead to a higher cost in most timeslots. Figure 5a illustrates the trace of the refilling cost during the considered horizon. The over-provisioning method has a higher cost than that of the DRLD in most of the timeslots. As shown in Algorithm 1, the system always explores the action space and exploits the knowledge at any given time step that is able to prevent the volumes of drugs from reaching bounded levels. On average, the DRLD can reduce the refilling cost by 12.31% compared to the over-provisioning method.

Figure 5b illustrates the refilling cost of the DRLD and ski-rental methods. Ski-rental is an online algorithm that makes decisions based on the current situation. However, the ski-rental method does not involve the learning phase to deal with changes in the system. Furthermore, the performance of the ski-rental method is sensitive to a competitive setting. In a complicated system with a large number of drugs and uncertain demands, the ski-rental algorithm does not present a suitable approach to solve DR2O. On average, the ski-rental method can obtain a better refilling cost than the use of over-provisioning. However, it is more unstable and results in costs that are on average 10.4% higher than those of the DRLD.

Finally, we evaluate the refilling cost by applying the max-min approach. The max-min inventory model seeks to reduce the gap between the max and min values, where the min value represents the reorder point and the max value represents the targeted quantity. Although this method is simple and non-optimized, it is able to provide an automation model for inventory management by using some triggers. Max-min is similar to over-provisioning, but more flexible if we set the lower and upper bounds to a high volume. The reorder quantity is calculated by the surplus between max and min. As shown in Figure 5c, this method flattens the filling cost during the time horizon. Its results indicate that it is a promising approach to control the shortage problem, but its cost is higher than that of the DRLD by 11.8%.

5.2.3. Evaluation of the Shortage Situation

We evaluate the shortage situation in a time horizon by applying four methods within a finite horizon: over-provisioning, DRLD, ski-rental, and max-min. During 30 timeslots, four-point-three-seven percent of the drugs hit a shortage level if applying ski-rental, while over-provisioning is slightly better with 4.09%. The max-min method has an even better result with 3.41% drugs at the shortage level. Our method, the DRLD, obtains the best result in this evaluation with only 2.21% of the drugs in a shortage situation in the considered timeslots. Figure 6 shows the shortage situations and their evolution for all four approaches. In detail, over-provisioning often hits serious shortage points, for example at Timeslots 14, 24, and 27 h, as the average value of the previous usage does not react well with the peak demand. The ski-rental method can obtain a better result compared to the over-provisioning method, but it also often reaches a shortage level. This method can react better to the changes in demand than the over-provisioning method, but it is too difficult to implement a correct setting of the *c*-competitive value for all the drugs in the system. The max-min method does not hit some peak points, but the shortage situation still occurs frequently during the evaluation period, similar to the over-provisioning method. As mentioned before, the intuition of the max-min approach is not that different than that of the over-provisioning method. Compared to all the baselines, the DRLD has the least amount of shortage points in the figure. At the peak shortage point (Time Slot 27), the DRLD has only 1.7% of its drugs that reach shortage, the lowest rate in the evaluation.

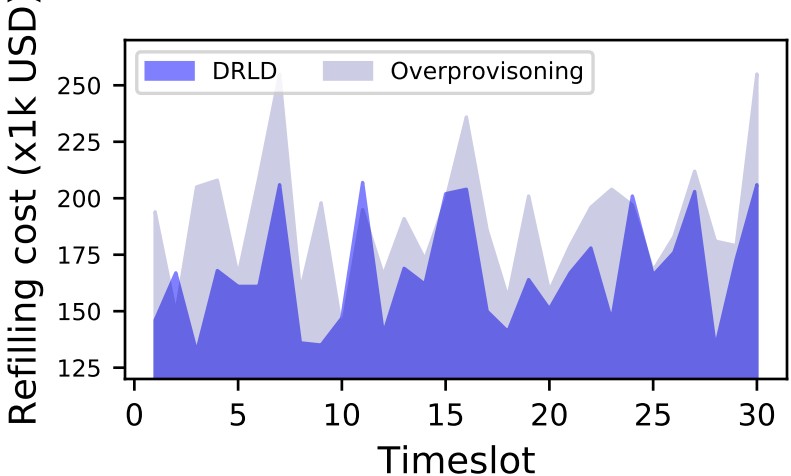

(**a**) DRLD vs. over-provisioning.

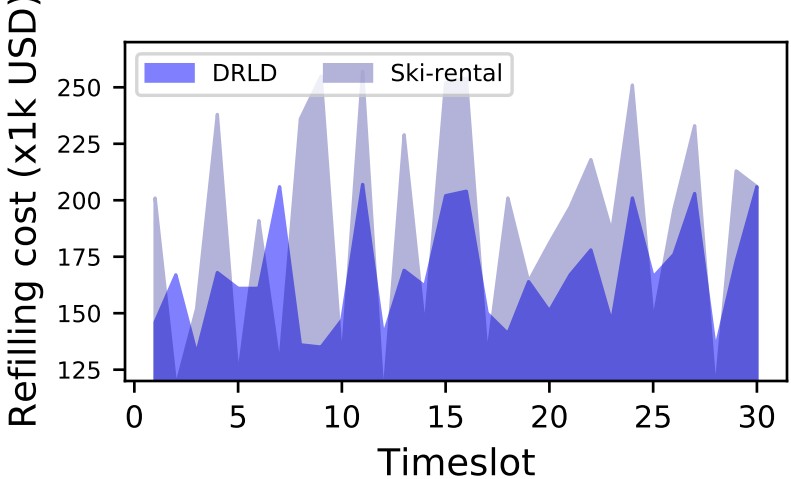

(**b**) DRLD vs. ski-rental.

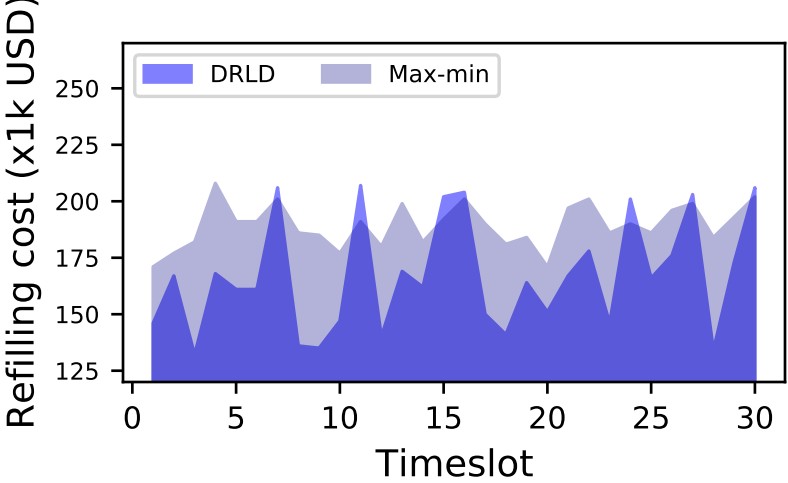

(**c**) DRLD vs. max-min.

**Figure 5.** Refilling cost evaluation.

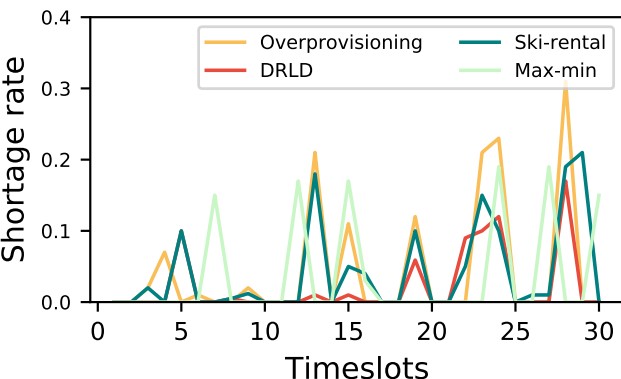

**Figure 6.** Shortage evaluation.

### 5.2.4. Evaluation of the Unexpected Rate

Finally, we evaluated the unexpected rates of drug refillings during 30 timeslots by using our proposed method and three baselines, as shown in Figure 7. We measured this aspect to illustrate the efficiency of our system for drug refillings and maintaining the volume of drugs at a stable level during a finite horizon. A high unexpected rate result is often considered as a low performance feature in inventory management. On average, over-provisioning has the highest rate by 1.675% and reaches the peak rate by 2.808%. It is agnostic of the user demand, and so, it is not able to prevent the system from a shortage or an overstock situation. If we increase the amount of the refilling volume, we can reduce the shortage, but it will proportionally increase the refilling cost and the number of redundant drugs. The ski-rental approach obtains an average rate of 1.54%, higher than that of the DRLD with 1.03%. This rate can be reduced by adjusting the $c$-competitive value. In practice, it is impossible to obtain the correct value for all drugs with an unknown usage demand. In Figure 7, the rate of the ski-rental approach fluctuates, with a large magnitude from the lowest rate of 0.1718% to the highest of 2.54%. With the awareness ability employed in the exploration and exploitation steps of the DRL, DRLD outperforms the other methods in this evaluation. The results illustrated in this figure reveal the efficiency of the DRLD at reducing the number of unexpected drugs in the refilling decision.

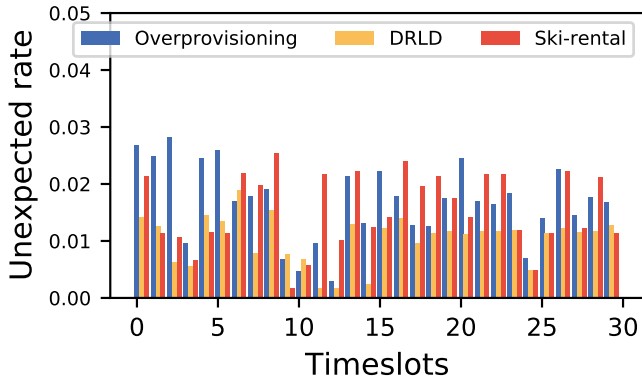

**Figure 7.** Unexpected rate evaluation.

## 6. Conclusions

As stated in the literature review, the hospital supply chain is responsible for ensuring that there is an adequate connection between hospitals, operations, and the revenue cycle. While inventory management is connected with the procedures of requesting, storing, and utilizing an institution's inventory, health care institutions across the globe are in

search of approaches to improve the efficiency of operations; i.e., effective inventory management that can reduce expenditures while in no way affecting medical care and services. An ineffective hospital supply chain leads to product shortages, product discontinuity, decreased patient safety, poor performance, distribution flaws, and technological mistakes that result in stock shortages in hospitals. Furthermore, our assessment reveals how health centers can effectively mitigate drug shortages in their hospital supply chains by adopting Machine Learning (ML). By promptly processing enormous volumes of data in order to discern patterns and uncover insights that are overly complicated for the human mind, ML can enable health care providers to consistently provide the right quantities, at the right cost, place, and time.

We propose a deep learning model to solve the hospital supply chain inventory control, using a mathematical programming model (DR2O) that can capture the requirements for drug refillings, while minimizing the purchasing and storing costs. To solve this optimization problem, we apply a deep reinforcement learning method that can determine how much of the volume of drugs should be refilled in each timeslot. The observations and analyses of the stock level changes are conducted based on intensive simulations. Our results outperform the other baselines with a finite horizon, specifically the over-provisioning, ski-rental, and max-min approaches. Our model is proven to be an efficient and promising mechanism with which to develop a dynamic programming framework for the management of hospital supply chain inventory.

**Author Contributions:** Conceptualization, T.A. Z.; Methodology, T.A.Z. and C.P.; Software, T.A.Z. and C.P.; Validation, T.A.Z. and C.P.; Investigation C.P.; Resources, T.A.Z., C.P., and Y.B.; writing—Original Draft Preparation, T.A.Z.; Writing—Review and Editing, T.A.Z., C.P., and Y.B.; Supervision, Y.B. All authors have read and agreed to the published version of the manuscript.

**Funding:** The authors would like to acknowledge NSERC for funding publications through discovery grant CRSNG RGPIN 2015-06253 .

**Institutional Review Board Statement:** Not applicable.

**Informed Consent Statement:** Not applicable.

**Data Availability Statement:** Not applicable.

**Conflicts of Interest:** The authors declare no conflict of interest.

**Sample Availability:** Not applicable.

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
