# Peer review of "Optimization of Inventory Management to Prevent Drug Shortages in the Hospital Supply Chain"

_applsci, doi:10.3390/app11062726_

Round 1

Reviewer 1 Report

  • The author needs to provide the justification of why a dynamic refilling drug optimization problem is an NP-hard problem.
  • The justification of DRL application needs to be further provided. Current argument is weak to say why DRL approach is appropriate to this problem in this situation.
  • What could be the technical contribution in DRL approach itself?
  • Detailed explanation is required for each benchmark approach and why each approach can be considered as a benchmark approach. There is no explanation with it.
  • There are several grammatical errors. Please ask the professional editing service to fix them. 

Author Response

Hello, Dear Respected Reviewer,

Please see the attached file concerning the response to your comments.

Thanks

Kind Regards

Reviewer 2 Report

This paper suggested very important problems in Hospital supply chain, Inventory control by refilling drugs. 

The proposed mathematical model and approach to find the near optimal solution are very interesting and show better performance. 

Whole script is well organized. However, I have found some limitations that should be clarified and explained in details. 

First of all, authors considered penalty cost from only shortage of inventory. But, generally, the penalty cost should include cost due to shortage as well as overstock. Like disposal cost can be included to the penalty cost. If authors think that it does not need to consider the penalty due to overstock, the reason should  be clarified or explained very well. 

Second, the explanation of simulation for comparing performance is not enough. I think authors should give more explanation about the performance evaluation, how to compare the performance, how better performance the proposed approach shows, and the reason why the proposed approach shows better performance.

In addition, the quality of graphics should be enhanced. Some information like legend and units need to be included.

Author Response

(The authors gave the same response as above.)

Reviewer 3 Report

The article is interesting.

The authors method is good.

Author Response

(The authors gave the same response as above.)
